# Effect of Deterpenated *Origanum majorana* L. Essential Oil on the Physicochemical and Biological Properties of Chitosan/β-Chitin Nanofibers Nanocomposite Films

**DOI:** 10.3390/polym13091507

**Published:** 2021-05-07

**Authors:** Rut Fernández-Marín, Muhammad Mujtaba, Demet Cansaran-Duman, Ghada Ben Salha, Mª Ángeles Andrés Sánchez, Jalel Labidi, Susana C. M. Fernandes

**Affiliations:** 1Environmental and Chemical Engineering Department, University of the Basque Country UPV/EHU, Plaza Europa 1, 20018 Donostia-San Sebastián, Spain; bensalhaghadaangle@gmail.com (G.B.S.); marian.andres@ehu.eus (M.Á.A.S.); jalel.labidi@ehu.eus (J.L.); 2Institute of Biotechnology, Ankara University, 06110 Ankara, Turkey or muhammadmujtaba443@gmail.com (M.M.); dcansaran@gmail.com (D.C.-D.); 3IPREM, CNRS, Universite de Pau et des Pays de l’Adour, E2S UPPA, 64000 Pau, France

**Keywords:** *Origanum majorana* L. essential oil, deterpenated fractions, beta-chitin nanofibers, chitosan, nanocomposite films, biological properties

## Abstract

Herein, the effect of three deterpenated fractions from *Origanum majorana* L. essential oil on the physicochemical, mechanical and biological properties of chitosan/β-chitin nanofibers-based nanocomposite films were investigated. In general, the incorporation of *Origanum majorana* L. original essential oil or its deterpenated fractions increases the opacity of the nanocomposite films and gives them a yellowish color. The water solubility decreases from 58% for chitosan/β-chitin nanofibers nanocomposite film to around 32% for the nanocomposite films modified with original essential oil or its deterpenated fractions. Regarding the thermal stability, no major changes were observed, and the mechanical properties decreased. Interestingly, data show differences on the biological properties of the materials depending on the incorporated deterpenated fraction of *Origanum majorana* L. essential oil. The nanocomposite films prepared with the deterpenated fractions with a high concentration of oxygenated terpene derivatives show the best antifungal activity against *Aspergillus niger*, with fungal growth inhibition of around 85.90%. Nonetheless, the only nanocomposite film that does not present cytotoxicity on the viability of L929 fibroblast cells after 48 and 72 h is the one prepared with the fraction presenting the higher terpenic hydrocarbon content (87.92%). These results suggest that the composition of the deterpenated fraction plays an important role in determining the biological properties of the nanocomposite films.

## 1. Introduction

Essential oils (EOs) are plants’ secondary metabolites, composed of a mixture of low molecular weight and volatile compounds that can be extracted from different parts of the plant including bark, roots, seeds, flower and leaves [1,2]. Nowadays, EOs have shown a particular interest in the development of bioactive materials because of their intrinsic properties namely antioxidant and antimicrobial activities [2,3].

EOs are mainly composed of terpenes and terpenoids, which are responsible for their biological activity, and are largely used in packaging, medicine, food and cosmetics sectors. The minor group of the EOs is terpenoids (oxygenated terpene derivatives). They are composed of alcohols, ketones and aldehydes, and are responsible for the organoleptic characteristics of essential oils. Terpenoids contribute to the protection of plants against insects, herbivores, fungal diseases and infestations [4,5]. On the other hand, the terpenes (simple hydrocarbons molecules) contribute to the flavor, fragrance, and color of plants. These compounds can easily degrade into undesirable compounds under the influence of light, heat, or oxygen, among others decreasing the quality and its market value. For this reason, it is important to carry out a deterpenation of the essential oils to separate the terpenes from the oxygenated terpene derivatives (terpenoids) to improve the stability and the bioactive properties of Eos [6]. Deterpenation has been done using different methods namely vacuum distillation, solvent extraction, membrane technologies, and extraction with ionic liquids [4,5,6,7,8,9,10].

Among the Eos, the one which is extracted from the *Origanum majorana* L. plant is one of the most promising due to its excellent antioxidant, antimicrobial, antifungal and antiparasitic properties and, consequently, high economic and industrial importance [10,11]. *Origanum majorana* L. belongs to the *Lamiaceae* family and is distributed in all of the Mediterranean region and Asia. It has been used since ancient times for food as a condiment and spice. Recently, Ben Salha and co-workers performed the deterpenation of the *Origanum majorana* L. EO and showed that the deterpenated fractions presented low antioxidant activity, but a good inhibition against *Aspergillus niger* [10].

To limit their instability and preserve their bioactivity, EOs and/or their fractions need to be encapsulated/incorporated in a matrix, in particular polymeric matrices for the development of multifunctional materials [3,12]. Chitosan-based matrices are good candidates for the incorporation of EOs as already demonstrated with Carum copticum, Thymus moroderi, Thymus piperella and Cinnamon verum essential oils [13,14,15]. Chitosan (poly-β-(1-4)-D-glucosamine, CS) is a cationic polymer obtained from the deacetylation of the chitin, which is mainly extracted from the exoskeleton of crustaceans. CS presents a great interest for the food industry and packaging application because of its intrinsic properties namely low toxicity, biocompatibility, biodegradability and bioactive activity [16,17]. However, as chitosan has limited mechanical properties, the incorporation of nanofibers like nanocellulose, lignocellulose nanofibers and chitin nanofibers and nanocrystals are often required to reinforce the matrix [16,18]. In our previous studies, we have shown the improvement of the biological and mechanical properties of bio-based matrices by the incorporation of nanochitin [3,19,20,21,22,23,24].

In this context, the objective of this work was to assess the effect of different *Origanum majorana* L. deterpenated oil fractions on the final properties, in particular antifungal activity and cell viability, on chitosan/β-chitin nanofibers nanocomposite films.

## 2. Materials and Methods

### 2.1. Materials

Clorhidric acid (HCl, 37% *w/w*, ACS reagent), ethyl acetate (HPLC grade), Tween 20, Ringer solution, 3-(4,5-dimethylthiazol- 2-yl)-2,5-diphenyltetrazolium bromide (MTT), DMEM medium, DPBS (Dulbecco’s Phosphate) and dimethyl sulfoxide (DMSO, 99.9 purity, ACS reagent) were purchased from Sigma-Aldrich (Madrid, Spain). Gallic acid monohydrate (extra pure) and Glycerol were obtained by Scharlau and PanReac (Barcelona, Spain), respectively. Potato dextrose agar (PDA) was provided by Merck (Kenilworth, NJ, USA). The penicillin/streptomycin was purchased from Biowest. The trypsinized and fetal bovine serum (FBS) were obtained from Biological Industries (Inc, Cromwell, CT, USA).

β-chitin powder (from squid pen) was supplied by Mahtani Chitosan PVT Ltd., Gujarat, India. β-chitin nanofibers (β-CHNF) were isolated in-house by acid hydrolysis using the microwave-assisted technique (Discover system, CEM, Matthews, NC, USA) under the following conditions: 1 M of HCl for 29.08 min at 79.08 W. The degree of acetylation was determined by ^13^C NMR [25] and was 88% (Appendix A). The nanofibers were analyzed by AFM and exhibited long and strongly tangled nanofiber morphology with lengths greater than 900 nm and average widths of 19.82 ± 1.16 nm (Appendix A).

Spider crab shells wastes were provided by a local restaurant (San Sebastián, Spain), and used to obtain chitosan as previously described [26] with slight modifications. First, chitin was extracted using three steps: the first step was deproteinization with 1M NaOH, in the second step calcium carbonate was removed by using 1 M HCl and, in the last step, lipids and pigments were removed by acetone following the procedure described by [23]. Then, chitin was deacetylated to obtain chitosan. For this purpose, chitin powder was treated with a NaOH solution (60% *w/v*) using a CH: NaOH ratio of 1:15 *w/v* at 130 °C in an oil bath for 4 h under N_2_ atmosphere. The obtained CS was filtered and washed. Finally, it was dried in an oven at 106 °C overnight. The degree of *N*-acetylation was determined by ^13^C-NMR and was 10%.

The essential oil (EO) and the 3 deterpenated fractions (F1, F2 and F3) of *Origanum majorana* L. (OM) were extracted by reduced pressure steam distillation as a function of boiling temperature. The terpenic hydrocarbon contents for F1, F2, F3 and OM were 87.92%, 12.90%, 8.98% and 50.70%, respectively. For total oxygenated terpenic derivatives the percentages of the samples were 12.05% (F1), 85.06% (F2), 88.86% (F3) and 47.36% (OM), as previously described [10]. 

### 2.2. Preparation of the Nanocomposite Films

CS (1% *w/v*) was dissolved in acetic acid solution (1% *v/v*) overnight with stirring, and the solution was then filtered to remove potential impurities. Afterwards, 0.16% *v/v* glycerol (CS-based) was added to chitosan solution as a plasticizer and the mixture was stirred for 5 min at 14,600 rpm at room temperature using an Ultra-Turrax (Heidolph Silent Crusher M., Schwabach, Germany). After that, Tween 20 (0.016% *v/v*, CS-based) was added as an emulsifier to assist essential oil dispersion in the film-forming solutions. Then, oil fractions (0.25% *v/v* F1, F2 and F3, CS-based) and essential oil (0.25% *v/v* OM, CS-based) were added and mixed at 14,600 rpm for 10 min. Finally, β-CHNF (0.5% *w/v*, CS-based) was added to each solution and mixed for 10 min at 14,600 rpm. All suspensions were degassed to remove the air bubbles. The films were prepared by the casting method overnight at 30 °C using 50-mm diameter Petri dishes. The nanocomposites were kept in a conditioning cabinet at 50 ± 5% relative humidity and 25 °C before use. A control film was prepared without the addition of *Origanum majorana* L. essential oil or deterpenated fractions. Table 1 shows the sample identification and composition.

### 2.3. Characterization of the Nanocomposite Films

#### 2.3.1. Physicochemical Characterization

##### Thickness

A digital micrometer (Ultra Präzision Messzeuge GmbH, Glattbach, Germany) was used to measure the thickness of the films. It was measured at six different random points on each sample with an accuracy of 0.001 mm. The thickness values were expressed in terms of the average of the measurement and standard deviation.

##### Moisture Content

The moisture content (MC) of nanocomposite films was measured by the method described by [3] with slight modifications. From each sample, 3 portions of 1.5 × 3 cm^2^ were cut, weighed and dried for 24 h at 106 °C in an oven (Memmert UN160 plus Twindisp, Büchenbach, Germany). The moisture content was calculated using the following equation:(1)$$MC\;\left(\%\right)=\frac{\left(W_{0}-W_{1}\right)}{W_{1}}\times 100$$
where *W*_0_ represents the initial weight of the sample (g) and *W*_1_ means the weight after 24 h drying in the oven (g). The MC was determined in triplicate and the average and its standard deviations were calculated for each sample.

##### Water Solubility

The water solubility (WS) was determined by the method performed by [27] with slight variations. Three portions of each film (1.5 × 3 cm^2^) were cut and dried in the oven at 106 °C (Memmert UN160 plus Twindisp, Büchenbach Germany) for 3 h until they reached constant weight. Each portion was immersed in 50 mL of distilled water. Then, they were kept immersed with stirring at 25 °C for 24 h. The undissolved portion was dried at 106 °C for 24 h and weighed. Solubility was calculated by the following Equation (2):(2)$$WS\;\left(\%\right)=\frac{\left(W_{0}-W_{1}\right)}{W_{0}}\times 10$$

##### Color of the Samples

The color of each nanocomposite film was measured using a colorimeter (PCE-CSM3, PCE, Spain) and the CIELAB color scale was used. In this scale, the parameters of the brightness (L*) were measured, going from L* = 0 (dark) to L* = 100 (bright), from −a* (greenish) +a* (reddish) and −b* (blue) to +b* (yellowish). A standard white plate was employed for the calibration of the colorimeter. The results were expressed by the parameters L*, a*, b* and the color difference was measured by the color change (∆E*). The calculations were made according to the following Equation (3):ΔE* = √[(ΔL*)2 + (Δa*)2 + (Δb*)2](3)

Color was measured in 10 random areas of each sample and the average values and their standard deviations were calculated.

##### Transmittance and Opacity

The transmittance (250–700 nm) and the opacity (600 nm) of the nanocomposite films were measured by a UV-Vis spectrophotometer (V-630 UV-Vis, Jasco, Pfungstadt, Germany). The opacity (Op) of each nanocomposite films was measured according to the following Equation (4):(4)$$Op=\frac{Abs_{600}}{x}$$
where *Abs*_600_ is the absorbance measured at 600 nm and *x* represents the thickness of the nanocomposite films (mm). The opacity was measured in triplicate for each film and the average values and their standard deviations were determined.

##### Water Contact Angle

The water contact angle was carried out with the OCA20 DataPhyscs (DataPhysics Instruments GmbH, Filderstadt, Germany) at room temperature. Each film surface was subjected to a drop of 4 μL distilled water. Images of the contact angle were obtained between 0 and 2 min using the SCA20 software. Six replicates were determined for each film.

##### Attenuated Total Reflection-Fourier Transform Infrared Radiation (ATR-FTIR)

Attenuated Total Reflection-Fourier Transform Infrared Radiation (ATR-FTIR) spectra were obtained employing a Spectrum Two FTIR Spectrometer with a Universal Attenuated Total Reflectance accessory (Perkin Elmer Inc., Waltham, MA, USA). The measurements were carried out using 64 scans and 4 cm^−1^ of resolution in the range of transmittance 600 to 4000 cm^−1^.

#### 2.3.2. Morphology

The morphology of the nanocomposite films was measured by scanning electron microscopy (SEM) (Hitachi Ltd. Japan). Before the analysis, the films were covered with 20 mm gold in a high vacuum. The samples were scanned at 10 kV acceleration voltage. The surface was analyzed with 2000×–10,000× as magnification and cross-section with 500×.

#### 2.3.3. Thermogravimetric Analysis (TGA) and Mechanical Properties

The thermal stability of the samples was carried out employing a TGA/SDTA 851 Mettler Toledo equipment (New Castle, DE, USA). All films were measured from room temperature to 750 °C at a heating rate of 10 °C/min under a continuous flow of nitrogen.

The mechanical properties of the films were carried out with an Instron 5967 tester machine (Instron, Norwood, MA, USA). For this, a cross head speed of 3 mm/min and a load cell of 500 N was used. The mean values of the tensile strength (TS), elongation and Young’s modulus (EM) were determined. The results represent the average of eight replicates of each sample (0.5 × 4.5 cm^2^).

#### 2.3.4. Antifungal Properties

The antifungal activity of the nanocomposite films was assessed against *Aspergillus niger* by employing the agar plate method following the Salaberria et al. 2015 approach with a slight modification [26]. Previously, *A. niger* was grown in a solid substrate of potato dextrose agar and incubated at 25 ± 2 °C for 72 h in sealed Petri dishes. After this time, an aliquot of spores was prepared in Ringer solution until a concentration of 1.29 × 106 cells/mL was obtained. The films (1 × 1 cm^2^ section) were then placed on the agar and aseptically inoculated with 40 μL of the *A. niger* suspension. Incubation was carried out for 1 week at 25 ± 2 °C and the number of colony-forming units per milliliter (CFU mL^−1^) was assessed. The fungal growth inhibition (FGI %) was calculated using the following Equation (5):(5)$$\;\mathrm{FGI}\;\%=\frac{C_{g}-T_{g}}{C_{g}}\times 100$$
where Cg is the average concentration in the control sample and Tg is the average concentration in the treated set. For each film, the test was performed in triplicate.

#### 2.3.5. Cytotoxicity Assay

L929 mouse fibroblast cell line was obtained from the Ministry of Health, Ankara, Turkey, and maintained in DMEM medium supplemented with 10% (*v/v*) fetal bovine serum 1% penicillin/streptomycin at 37 °C in a humidified incubator in an atmosphere of 5% CO_2_. L929 cells were cultured as a 70% to 80% confluence and cells were harvested after being trypsinized. The images of the produced fibroblast cells are displayed in the Appendix A). In vitro assay was performed to determine the cytotoxic effects of nanocomposite films. The samples were sterilized by placing them under UV light for 40 min. Cells were seeded into a 96-well plate at a density of 1 × 10^4^ cells and incubated for 24 h at 37 °C. After the 24 h incubation, the medium was aspired out and L929 cells were supplied with a fresh medium. Then, the cells were treated with nanocomposite film samples at a concentration of 8 mg/mL and incubated for 24, 48 and 72 h. DMEM was selected as a negative control. The cell viability was evaluated by following the MTT (3- (4, 5-dimethylthiazol-2-yl)-2, 5-diphenyltetrazolium bromide) reduction assay. An amount of 20 μL MTT solution (5 mg/mL of stock in DPBS) was added into each well and then incubated for 4 h at 37 °C in an incubator. The culture medium was removed and 100 μL DMSO was added into each well to extract the insoluble formazan crystals within the cells. The plates were shaken for 15 min. The presence of viable cells was demonstrated by purple color due to the formation of formazan crystals. The absorbance was measured at 540 nm using a microplate reader. The results represent the average values of four experiments.
(6)$$\mathrm{Viability}\;\left(\%\right)=\frac{X_{\mathrm{OD}}}{X_{{\mathrm{OD}}_{0}}}\;\times \;100$$
where X_OD_ represents the mean of optical density of treated cells and X_OD_0__ is the mean of optical density of control cells
Cytotoxicity (%) = Percent of viability × 100(7)

#### 2.3.6. Statistical Analysis

The statistical analysis was carried out using a one-way analysis of variance (ANOVA) by SPSS Statistical software (Version 24, Inc. Chicago, IL, USA). The significant difference values were calculated by Duncan’s multiple range test. Results are given as mean ± standard deviation and *p* values < 0.05 were statistically significant.

## 3. Results and Discussion

### 3.1. Physicochemical Characterization

Figure 1 shows the general aspect of the nanocomposite films. All samples are homogeneous, translucent and with a slightly yellowish tone. The incorporation of the OM and its fractions slightly affected the thickness of the films (*p* < 0.05), as listed in Table 2. This could be due to the increase in the density as a result of the formation of intermolecular interactions between β-CHNF, CS and essential oil [28].

#### 3.1.1. Moisture Content and Water Solubility

The effect of the incorporation of the deterpenated fractions of *Origanum majorana* L. essential oil on the moisture content, solubility and hydrophobicity of chitosan/β-chitin nanofibers nanocomposite films were also assessed. The moisture content of the nanocomposite films is listed in Table 2. As expected, the addition of the essential oil resulted in a decrease in the moisture content, and no significant differences were observed between the films with fractions or essential oil (*p* > 0.05). Similar results were also observed by Fernández-Marín et al., who added oregano essential oil and chitin nanocrystals to poly(vinyl alcohol) films [3], and other authors demonstrated this tendency when they combined chitosan, olive oil and cellulose nanocrystals [29]. Regarding the water solubility behavior of the samples, it was observed that the addition of the OM and its fractions to the CS/chitin nanofibers matrix decreased the water solubility (Table 2). As for the moisture content, it was observed that films containing essential oils did not show significant differences between them (*p* > 0.05).

#### 3.1.2. Color Properties

The color parameters of the nanocomposite films were also assessed; the data listed in Table 3 show an L* (lightness/darkness) of around 90 for all samples. Similar results were obtained by Bonilla et al. using chitosan/gelatin as matrix and eugenol and ginger oil [30]. The a* parameter of all films was quite similar with a value of around one and with a positive sign, indicating a tendency toward reddish color (*p* < 0.05). The yellowness effect given by parameter b* was also very similar for all the films, except for the CSNF-F1 films (11.02 ± 0.51) (*p* < 0.05). The yellowish color is attributed to the color of the oils themselves [30,31]. Consequently, the total color (ΔE) of the films was different for the CSNF-F1 film, resulting in a film with more color. In general, the films with OM and its fractions showed higher values of total color (*p* < 0.05). This tendency was also observed in previous studies using chitosan films with eucalyptus extract [32] and with apple peel extracts [33].

#### 3.1.3. Optical Properties

The opacity of the nanocomposite films was calculated by using Equation (2) taking into account the absorbance at 600 nm and the thickness (Table 3). As expected, the lowest opacity value was observed for the CSNF film with a value of 4.76 ± 0.42. The opacity increased with the incorporation of OM and its fractions. The nanocomposite films with the deterpenated fractions showed the highest opacity (7.47 ± 0.77 for CSNF-F2 and 7.90 ± 0.35 for CSNF-F3). These values were observed in accordance with the transmittance of the films (Figure 2). Notably, it was observed that films with essential oils showed higher opacity [3,30,34].

The transmittance of the films was assessed from 700 to 250 nm (Figure 2). The films with OM or fractions presented transmittances around 35 and 50% in the visible light region. Interestingly, in the ultraviolet region, the transmittance of all films was recorded as below 20% (350–250 nm), and CSNF-F1 nanocomposite film blocks UV light in the range of 300–250 nm. CSNF-F1 nanocomposite film presents a higher percentage of terpenes. Similar results were reported by Sahraee et al. 2017, who employed corn oil in gelatin films incorporated with nanochitin [27]. Wu et al. 2014 also showed that the rising concentration of oregano oil in chitosan films decreased the transmittance almost completely in the UV range [35].

#### 3.1.4. ATR-FTIR

The chemical structure of the nanocomposite films was analyzed by ATR-FTIR (Figure 3). All nanocomposite films exhibited the typical chitosan bands at 1640 and 1550 cm^−1^ corresponding to amide I and amide II, respectively [36,37,38]. The band around 1405 cm^−1^ was observed in all films and was attributed to C–H stretching vibration [39]. A difference was observed in the band appearing at 1377 cm^−1^ in the films CSNF-F1, CSNF-F2, CSNF-F3, CSNF-OM, which was attributed to the C–N stretching of the bioactive compounds of the essential oils [40,41]. Another remarkable difference was noticed between the films with and without oil. In CSNF films, two bands appeared around 1065 and 1021 cm^−1^ which were assigned to C–O bending secondary and primary O–H groups, respectively [38]. While in the films with oil and fractions a single band appeared around 1030 cm^−1^ corresponding to C–OH stretching vibration [41].

#### 3.1.5. Water Contact Angle

The hydrophilic/hydrophobic behavior of nanocomposite films was studied using the water contact angle technique. In general, angles above 90° indicate a hydrophobic surface while angles below 60° indicate a hydrophilic surface [42,43]. Figure 4 shows the contact angle values for all samples for 0 and 120 s. As expected, the CSNF film demonstrated a hydrophilic behavior, with an initial contact angle of around 85°, which were attributed to the surface rigidity of the film (Figure 5), that quickly fell to characteristic values of chitin and chitosan (54.9° at 120 s, Figure 4) [44]. The incorporation of OM essential oil and its fractions showed a different behavior; the starting contact angles were around 80° and the drop was maintained stable over time (from 0 to 120 s, Figure 4). This fact is due to the presence of polyphenolic groups in the oils [17,43,45]. For instance, CSNF-F1 and CSNF-OM films demonstrated a contact angle loss of 1.6 and 1.15°, respectively, showing the effect of the hydrophobic behavior of the oils.

### 3.2. Morphology

Scanning electron microscope (SEM) analysis was employed to assess the morphology of the nanocomposite films. Figure 5 shows the cross-section and the surface images of all nanocomposite films. The presence of the chitin nanofibers in all samples was confirmed by both the cross-section and surface SEM images (Figure 5A–E). In general, the nanocomposite films exhibited a homogeneous morphology with a rough surface. As shown by the cross-section images, the addition of the deterpenated fractions of *Origanum majorana* L. essential oil promoted more compact (dense) films, probably resulting in the establishment of strong interactions between the EO molecules and the chitosan chains and chitin nanofibers. The presence of dots at the surface of the matrices with deterpenated fractions of *Origanum majorana* L. essential oil was also seen (Figure 5C,D), as observed previously by Valizadeh et al. after the incorporation of EOs on chitosan and carboxymethyl cellulose matrices which could be due to emulsified drops of essential oil [46]. These results are also consistent with the results obtained by Khan et al. in their cellulose nanocrystals reinforced chitosan films [47].

### 3.3. Thermogravimetric Analysis and Mechanical Properties

The influence of the introduction of the deterpenated fractions of *Origanum majorana* L. essential oil on the thermal stability and mechanical properties of chitosan/β-chitin nanofibers nanocomposite films were also investigated by TGA and tension tests, respectively.

The onset thermal decomposition and the temperature of a maximum of mass loss were determined from TGA and derivative TGA (dTGA) curves (Figure 6). TGA and dTGA of the CSNF sample displayed five main degradation steps. The first step occurred bellow 100 °C and was attributed to the acetic acid and water evaporation. In the second step the onset temperature was 150 °C with a maximum degradation at 200 °C and was assigned to the decomposition of the major of Tween 20 molecules and loss of some glycerol molecules. The maximum degradation temperatures at 330 and 350 °C (third and fourth steps) were attributed to chitosan and nanochitin degradation, respectively [48,49,50]. The small decomposition step between 425 and 500 °C was ascribed to the remaining Tween 20 molecules [48,50]. As can be observed by the TGA and dTGA curves, the main difference between samples containing the EO fractions and the CSNF matrix was observed in the second degradation step. This step is larger and presents the main loss of compounds. The onset temperature starts at around 150 °C and the maximum degradation temperature was observed at 240 °C. This consequent loss was mainly attributed to the loss of the low molecular weight molecules of the EOs, and also to the Tween 20 and glycerol molecules loss. Interestingly, the CSNF-F1 film was shown to be the least thermostable material. The second degradation step displayed an onset temperature of 100 °C and a maximum degradation temperature of around 150 °C. The maximum degradation temperatures of chitosan and nanochitin were 250 and 300 °C, respectively.

The results of the mechanical properties (tensile strength, Young‘s Modulus and Elongation) are shown in Table 4. In general, the incorporation of OM and its different fractions on the CSNF matrix caused a significant decrease in the tensile strength (TS and YM) (*p* < 0.05). This trend was also observed by Ardekani et al. when incorporating 10% *Zataria multiflora* oil and mats nanofibers in a chitosan and poly(vinyl alcohol) matrix [51]. Mohammadi et al. obtained similar results in their whey protein films with cinnamon oil and chitosan nanofibers [52]. The elongation (E %) was also affected by the addition of OM oil and its fractions. These values increased significantly (*p* < 0.05) in CSNF-F1, CSNF-F2, CSNF-F3 and CSNF-OM films. Essential oils act as plasticizers because they reduce the intermolecular forces in the chitosan network allowing mobility in the chains and enhancing the flexibility of the film [30,33,53].

### 3.4. Antifungal Properties

The effect of the incorporation of the different fractions and OM essential oil in the CSNF films on the antifungal activity against *Aspergillus niger* was analyzed (Figure 7). Photographs representing the general aspect after 7 days of incubation are shown in the Appendix A. This fungus was chosen because is one of the main contaminants of food, especially fruits and vegetables, thus important for packaging applications.

As already reported by Salaberria et al., the nanocomposites prepared with nanochitin present significant inhibitory activity in CS matrices (FGI % around 64.4%) [26] and concentration of CFU. mL^−1^ was below 60,0000 (Figure 7). Moreover, when OM essential oil or its fractions were added into the nanocomposite films an increase in the antifungal activity was observed (83.9, 72.7, 85.5 and 85.9% for CSNF-OM, CSNF-F1, CSNF-F2 and CSNF-F3, respectively). The nanocomposite films prepared with the deterpenated fractions CSNF-F2 and CSNF-F3 exhibited higher FGI % than the other oil-based films. The major compounds in the fractions of these films are oxygenated terpenes derivatives (terpenoids) consisting of polar terpenes and alcohols, which have been shown to have a negative effect on fungal growth, as previously demonstrated [10,54]. CSNF-OM also showed good activity against *Aspergillus niger*. As expected, CSNF-F1 nanocomposite films (FGI of around 72%) presented a lower activity, because its major compounds were hydrocarbonated terpenes such as γ-terpinene, α-terpinene or β-pinene, which are usually less effective against *A. niger*, as previously verified [10,55].

The mechanism of the oils against fungi could be justified by the penetration or interaction of these compounds in the cell membrane affecting the fungi functions, like respiratory inhibition and thus causing cell death [3,48].

### 3.5. Cytotoxicity Assay

The cell viability of the nanocomposite films on the growth of L929 murine fibroblast cells was analyzed by MTT (3- (4, 5-dimethylthiazol-2-yl)-2, 5-diphenyltetrazolium bromide) assay at different incubation times (24, 48 and 72 h, Figure 8). According to the standard method ISO 10993-5, the samples with a cell viability ratio below 70% are considered cytotoxic. As expected, the CSNF nanocomposite films are non-cytotoxic, as previously demonstrated [19,20]. The analyses carried out to assess the cytotoxic activity of the obtained nanocomposite films prepared with the different EO fractions showed that these extracts have different effects on the tested cell line. Interestingly, CSNF-F1 nanocomposite films, revealed no toxic effects on the viability of L929 fibroblast cells after 48 and 72 h. On the other hand, the film samples CSNF-F2, CSNF-F3, and CSNF-OM showed cytotoxic effects on the viability of L929 fibroblast cells. This can be ascribed to the high concentration of terpenoids molecules of the deterpenated *Origanum majorana* L. essential oil that may cause internal cell damage leading to cell death [56]. Additionally, the samples may have the potential to damage the cell membrane after interaction with intracellular substances inside the cell.

## 4. Conclusions

In this work, the effect of three deterpenated *Origanum majorana* L. essential oil fractions on the physicochemical, mechanical and biological properties of chitosan/β-chitin nanofibers nanocomposite films was investigated. The presence of the original essential oil or of its deterpenated fractions increased the opacity, gave them a yellowish color and improved the hydrophobicity of the nanocomposite films.

The data showed that the proportion of terpenes hydrocarbons or oxygenated terpenes derivatives in the oil fractions influences the biological properties of the materials. The nanocomposite films prepared with oxygenated terpene derivatives showed the best antifungal activity against *Aspergillus niger,* presenting a FGI of 85.49% for CSNF-F2 and of 85.90% for CSNF-F3. Nonetheless, these materials presented cytotoxic properties on the studied L929 fibroblast cells, with a cell viability inferior at 30%. On the other hand, the nanocomposite films prepared with terpenes hydrocarbons (CSNF-F1), showed no cytotoxicity on the viability of L929 fibroblast cells after 48 and 72 h, with a cell viability of around 90%. Therefore, these findings suggest that these nanocomposite films present a good potential for packaging or medical and pharmaceutical sectors.

## Figures and Tables

**Figure 1 polymers-13-01507-f001:**
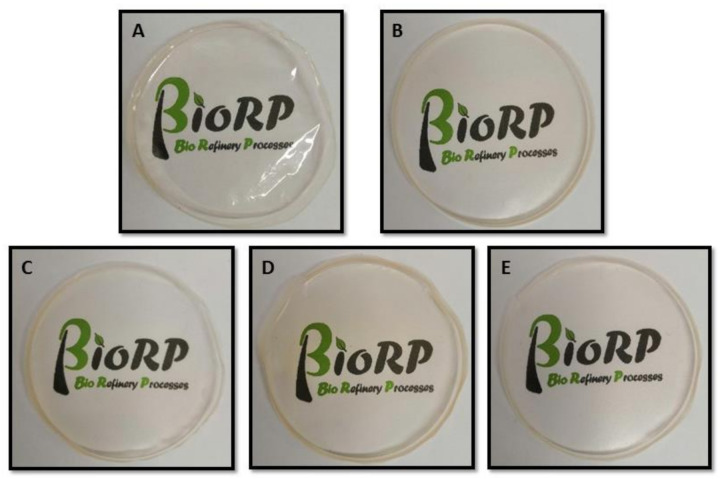
The general appearance of the films. (**A**) CSNF; (**B**) CSNF-F1; (**C**) CSNF-F2; (**D**) CSNF-F3; (**E**) CSNF-OM.

**Figure 2 polymers-13-01507-f002:**
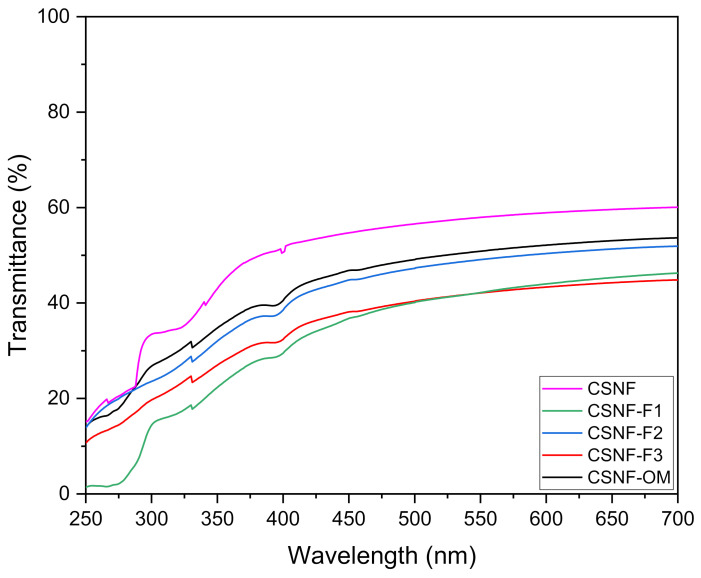
UV-Vis spectra (700–250 nm) of the nanocomposites films.

**Figure 3 polymers-13-01507-f003:**
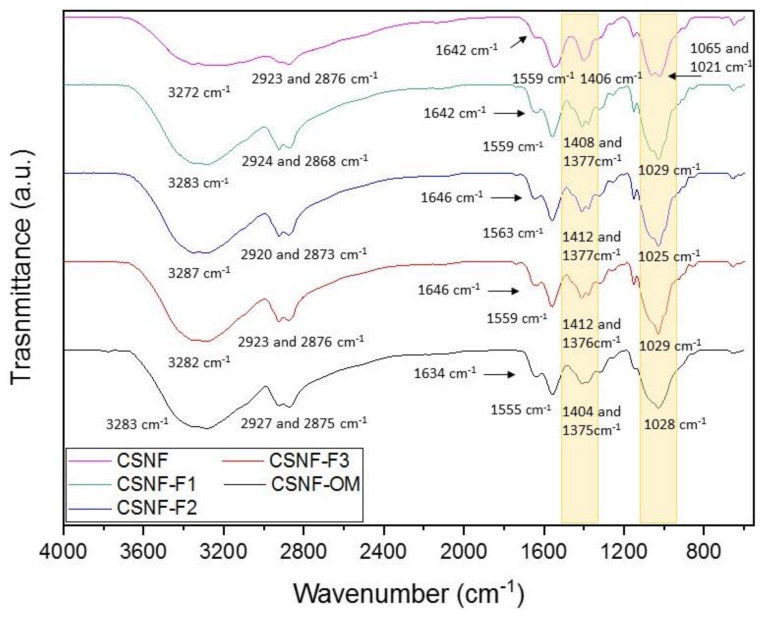
ATR-FTIR spectra of the nanocomposite films.

**Figure 4 polymers-13-01507-f004:**
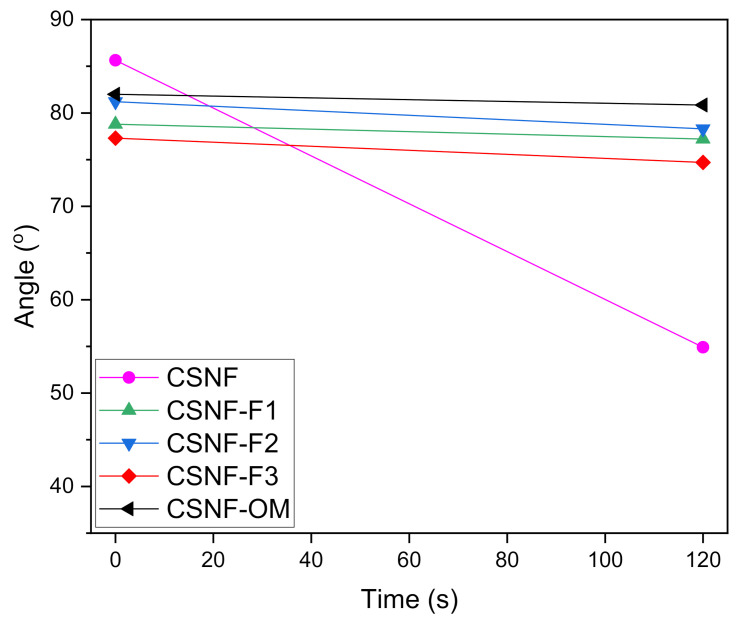
The water contact angle of nanocomposite films.

**Figure 5 polymers-13-01507-f005:**
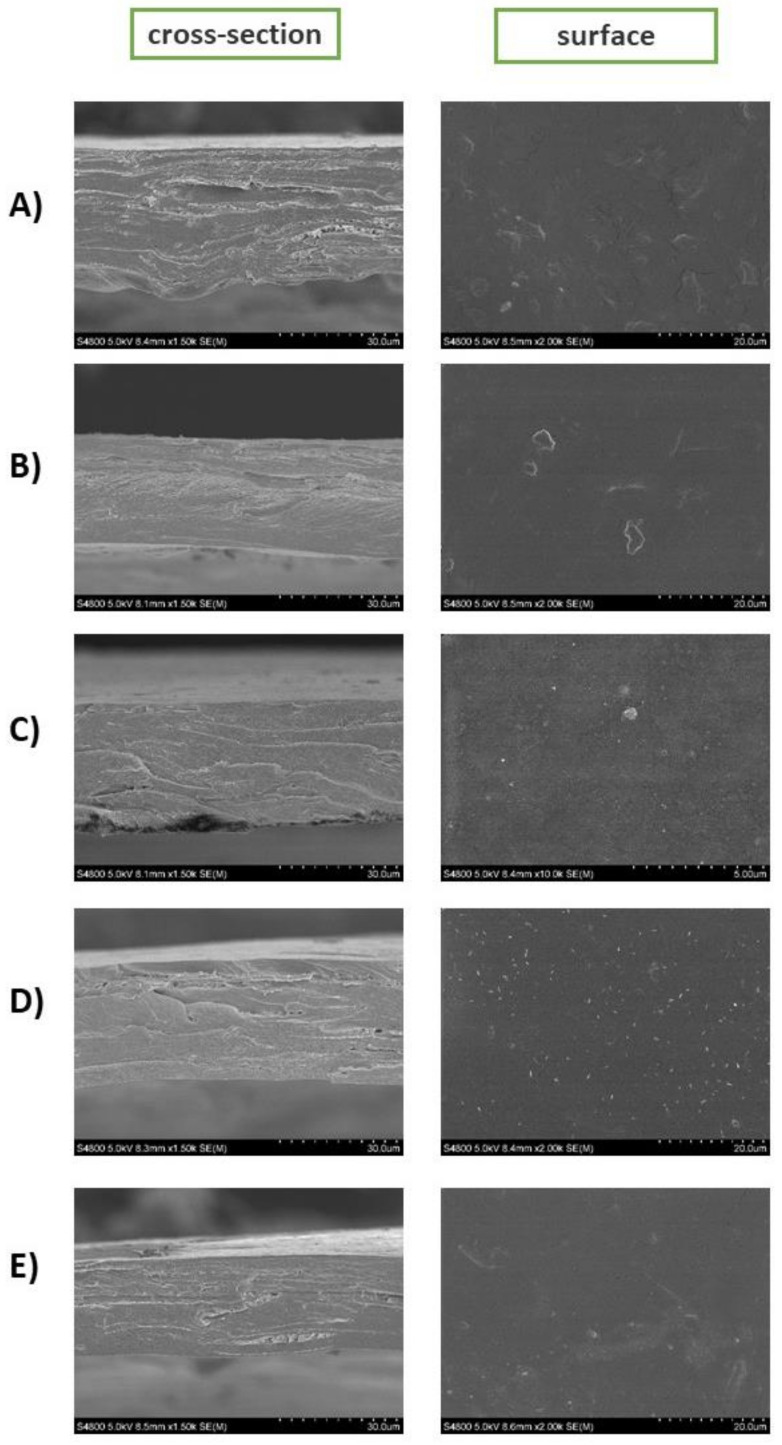
Cross-section and surface images of scanning electron microscopy of (**A**) CSNF; (**B**) CSNF-F1; (**C**) CSNF-F2; (**D**) CSNF-F3; (**E**) CSNF-OM films.

**Figure 6 polymers-13-01507-f006:**
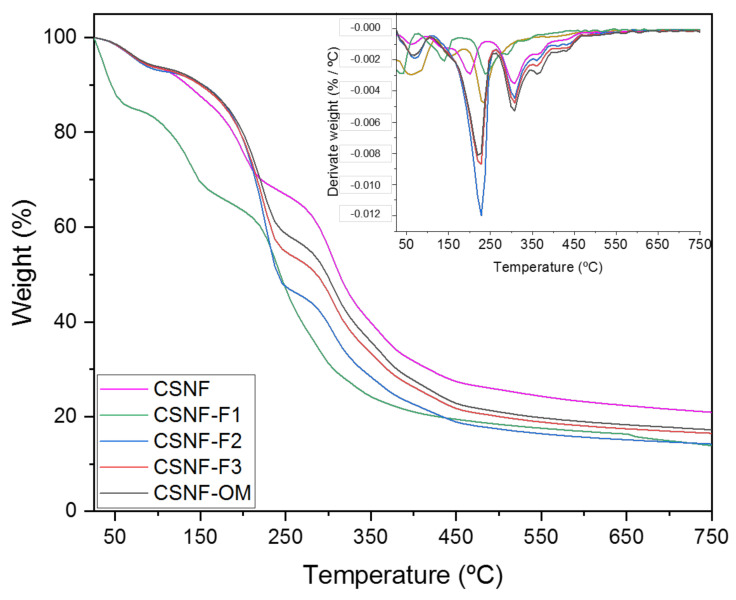
TGA and dTGA curves of the nanocomposite films.

**Figure 7 polymers-13-01507-f007:**
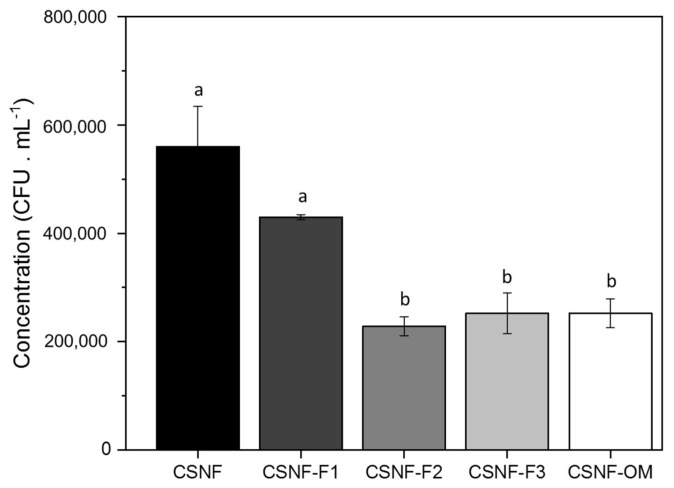
Effect of CSNF, CSNF-F1, CSNF-F2, CSNF-F3 and CSNF-OM nanocomposite films on the growth of Aspergillus niger. The error bar represents the standard deviation (*n* = 3). Superscript letters in the bars indicate significant differences among the films (Duncan´s test, *p* < 0.05).

**Figure 8 polymers-13-01507-f008:**
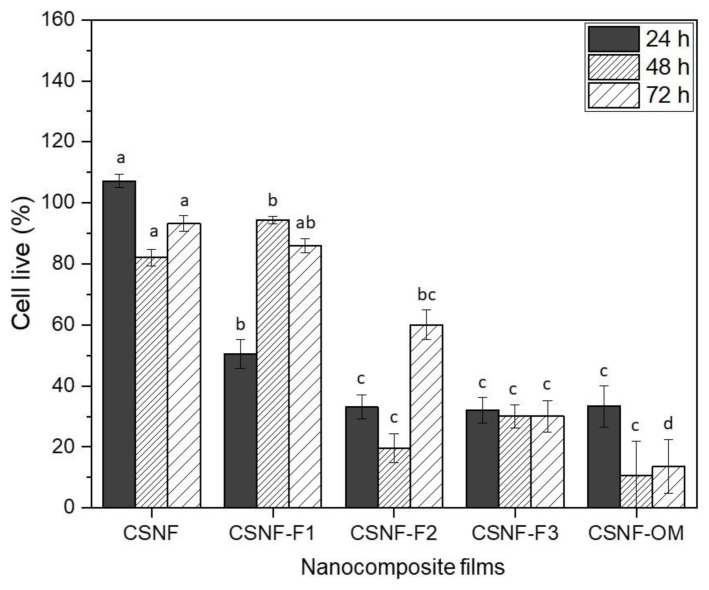
Cell viability of the nanocomposite films. The error bar corresponds to standard deviation (SD, *n* = 4). Letters in the bars with the same color denote significant differences among the different films and the same cell culture time (Duncan’s test, *p* < 0.05).

**Table 1 polymers-13-01507-t001:** Identification and composition of the nanocomposite films.

Samples	Samples Identification	β-CHNF (% *w/v*) *	Fractions and Essential Oil (% *v/v*) *
**Chitosan + β-CHNF**	CSNF	0.5	-
**Chitosan + β-CHNF + F1**	CSNF-F1	0.5	0.25
**Chitosan + β-CHNF + F2**	CSNF-F2	0.5	0.25
**Chitosan + β-CHNF + F3**	CSNF-F3	0.5	0.25
**Chitosan + β-CHNF +OM**	CSNF-OM	0.5	0.25

* % *v/v* and % *w/v*, chitosan (CS) based.

**Table 2 polymers-13-01507-t002:** Thickness, moisture content and water solubility of the nanocomposite films.

Samples	Thickness (μm)	Moisture Content %	Water Solubility %
**CSNF**	41.33 ± 1.97 ^a^	54.98 ± 2.48 ^a^	57.87 ± 4.78 ^a^
**CSNF-F1**	45.00 ± 3.03 ^b^	40.41 ± 7.96 ^b^	32.53 ± 1.30 ^b^
**CSNF-F2**	41.83 ± 5.34 ^a^	45.20 ± 9.03 ^b^	34.56 ± 3.07 ^b^
**CSNF-F3**	44.33 ± 4.41 ^b,c^	53.79 ± 1.48 ^b^	49.27 ± 3.60 ^b^
**CSNF-OM**	42.83 ± 5.19 ^c^	42.72 ± 7.45 ^b^	32.37 ± 4.87 ^b^

The values are averages ± standard deviation (thickness *n* = 6; moisture content, water solubility *n* = 3). Different letters in the same column depict significant differences between samples (Duncan‘s test, *p* < 0.05).

**Table 3 polymers-13-01507-t003:** Color and opacity of nanocomposite films.

Samples	L*	a*	b*	ΔE	Opacity
**CSNF**	91.83 ± 0.69 ^a^	1.23 ± 0.11 ^a^	6.02 ± 0.69 ^a^	1.73 ± 0.56 ^a^	4.76 ± 0.42 ^a^
**CSNF-F1**	90.20 ± 0.23 ^a^	1.35 ± 0.12 ^a^	11.02 ± 0.51 ^b^	6.68 ± 1.05 ^b^	7.08 ± 0.78 ^b^
**CSNF-F2**	91.95 ± 0.47 ^b^	0.91 ± 0.06 ^b^	6.84 ± 1.07 ^a^	2.21 ± 1.05 ^a^	7.47 ± 0.77 ^b^
**CSNF-F3**	91.42 ± 1.12 ^a^	1.23 ± 0.36 ^a^	7.13 ± 2.09 ^a^	2.85 ± 2.21 ^a^	7.90 ± 0.35 ^b^
**CSNF-OM**	91.97 ± 0.33 ^b^	0.94 ± 0.05 ^b^	6.56 ± 0.83 ^a^	1.94 ± 0.83 ^a^	6.61 ± 0.53 ^b^

The values were mean ± standard deviation (L*, a*, b* and ΔE *n* = 10; opacity *n* = 3). Different letters in the same column indicate significant differences between nanocomposite films (Duncan‘s test, *p* < 0.05).

**Table 4 polymers-13-01507-t004:** Mechanical properties of the films (TS: tensile strength; YM: Young’s modulus; E: elongation).

Samples	TS (MPa)	YM (MPa)	E %
**CSNF**	15.25 ± 1.86 ^a^	328.25 ± 18.67 ^a^	44.87 ± 8.12 ^a^
**CSNF-F1**	17.88 ± 5.32 ^b^	310.98 ± 61.05 ^b^	64.40 ± 14.64 ^b^
**CSNF-F2**	11.52 ± 3.97 ^b,c^	77.09 ± 11.77 ^c^	67.92 ± 33.13 ^b^
**CSNF-F3**	12.20 ± 3.56 ^c^	153.39 ± 48.92 ^d^	46.90 ± 15.01 ^a,c^
**CSNF-OM**	11.49 ± 3.27 ^b,c^	133.89 ± 33.38 ^d^	50.66 ± 12.62 ^b,c^

The values represent average ± standard deviation (mechanical properties, *n* = 8). Superscript letters in the same column indicate significant differences among the films (Duncan’s test, *p* < 0.05).

## Data Availability

Not applicable.

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
