# Peer review of "Effect of Deterpenated Origanum majorana L. Essential Oil on the Physicochemical and Biological Properties of Chitosan/β-Chitin Nanofibers Nanocomposite Films"

_polymers, 2021, doi:10.3390/polym13091507_

Round 1
Reviewer 1 Report
The paper <The effect of deterpenated fractions of Origanum majorana L. essential oil on the final properties of chitosan/ β-chitin nano-3 fibres nanocomposite films> is an interesting one and could be published in Polymers after minor revision.
There some minor correction listed below which must be done.
Abstract-first line terpenoids instead of terpenoid
Line 53- ionic liquid. I suggest ionic liquids, because coud be more than one
Line 57- Other authors employed ionized liquids- I suggest the form ionic liquids
Line 63 in all Mediterranean and Asia- I suggest Mediterean area or region
Line 113-118 There are no information about chemical analyses of different fractions. The authors have cited one of their work published in Ind. Crops Prod. A brief description of the analysis I consider that is also necessary.
Line 165-x instead of X
Line 228 -1.29 x 106 spores/mL is not correct
Line 504- Origanum majorana L. fractions – I suggest Origanum majorana L. oil fractions
Line 507 -The addition of the essential oil or its deterpenated fractions/ I suggest:
The addition of the essential oil or of its deterpenated fractions
Author Response
Reviewer 1
Comments and Suggestions for Authors
The paper <The effect of deterpenated fractions of Origanum majorana L. essential oil on the final properties of chitosan/ β-chitin nano-3 fibres nanocomposite films> is an interesting one and could be published in Polymers after minor revision.
There some minor correction listed below which must be done.
Abstract-first line terpenoids instead of terpenoid
Line 53- ionic liquid. I suggest ionic liquids, because coud be more than one
Line 57- Other authors employed ionized liquids- I suggest the form ionic liquids
Line 63 in all Mediterranean and Asia- I suggest Mediterean area or region
Line 113-118. There are no information about chemical analyses of different fractions. The authors have cited one of their work published in Ind. Crops Prod. A brief description of the analysis I consider that is also necessary.
Line 165-x instead of X
Line 228 -1.29 x 106 spores/mL is not correct
Line 504- Origanum majorana L. fractions – I suggest Origanum majorana L. oil fractions
Line 507 -The addition of the essential oil or its deterpenated fractions/ I suggest:
The addition of the essential oil or of its deterpenated fractions
We thank Reviewer 1 for the suggestions given to improve our manuscript. We have done all the proposed modifications.
Reviewer 2 Report
Find the reviewer´s comments in the attached file.

Author Response
Reviewer 2
Remarks for the Author (Polymers- 1204339-peer-review-v1)
This manuscript contains some interesting findings. However, the manuscript is not written clearly, and the results were not discussed properly. There are many problems with the manuscript as it stands (detailed below) and these need to be addressed before it can be considered further. I thus recommend the paper be reconsidered after major revisions.
- The title of the manuscript is not opt, it should be modified
We have changed the title as following:
Effect of deterpenated Origanum majorana L. essential oil on the physicochemical and biological properties of chitosan/ β-chitin nanofibres nanocomposite films
- The abstract is tediously long it should be shortened with proper representation
- Abstract: Rewrite the lines 25-27.
We thank Reviewer 2 for this suggestion. In response to suggestions 2 and 3, we have improved the abstract.
- The introduction section is tediously long and should be revised entirely so that the reader can clearly identify the scientific problems solved by this research.
We agree with the reviewer's comments. We have improved the introduction section.
- UV-Vis spectra of composite materials are not clear and it does not look like UV-vis spectra. Therefore, the authors should report the complete UV spectra (100-800 nm) of materials.
Thanks for your suggestion; we have improved the clarity of the image to look like UV-vis. In this work, the scan was carried out between 250-700 nm in transmittance mode, which is adequate to do the necessaries conclusions about the transmittance of the films in the visible region and UV.
- More experiments should be needed to support the structural evidence of key materials.
We thank the reviewer for this suggestion. However, in our opinion, in the present work, ATR-FTIR and SEM are sufficient for the structural evidence of the materials. We will be tankful if the reviewer suggests us the missing experiments in order to consider it in our future works.
- FTIR peak values should be added to the FTIR spectra of materials.
Thanks for the suggestion we have now added the FTIR peaks in Figure 3.
- How did authors obtain fibroblast cells? The authors could add some cell images in the
Manuscript
Thank you for the comment. Detailed regarding the cells attainment has already been given in the methodology section. We have now included the images of the produced fibroblast cells in Supplementary material as Figure S4.
- Need more detailed description of cell viability, the authors should provide more
information on viability and toxic effect of CSNF-F2, CSNF-F3, and CSNF-OM
In order to clarify the comment, we have now included a better explanation to the section 3.5 and a reference was added:
Patil Shriniwas P, Kumbhar Subhash T. Antioxidant, antibacterial and cytotoxic potential of silver nanoparticles synthesized using terpenes rich extract of Lantana camara L. Leaves. Biochemistry and Biophysics Reports 10 (2017) 76–81
- Can the authors experimentally prove the preservation of essential oil after integration with
-chitin nanofibers composites?
Thank you for your interesting question but we have not yet performed this analysis. However, with the experiments carried out in this work we have been able to demonstrate the integration of the essential oil and its fractions in our materials.
- The following reference should be cited in the article. Because in the following article
PEEK/CNOs fibers nanocomposite films were prepared and physicochemical properties
were investigated significantly. The obtained composite fiber films displayed augmented
mechanical, thermal, and cell viability properties. Therefore, the authors should compare
their results, specifically mechanical properties with the following article. Which will
enrich the quality of the current manuscript.
Pharmaceutics 2020, 12(12), 1208; https://doi.org/10.3390/pharmaceutics12121208
It would be more realistic to cover such kind of research work in the current manuscript.
Thank you very much for recommending the article. Nonetheless, after reading it we do not know if there is a mistake because it deals with the design and synthesis of poly (N-(4-aminophenyl) methacrylamide))-carbon as nanofiller to develop layer-by-layer thin films with anilinated-poly (ether ether ketone). Therefore, we believe that it does not match with our work and that is why we did not consider it appropriate to cite it in the end. In case it is a mistake, we would like you to indicate the right paper.
Reviewer 3 Report
polymers-1204339
The effect of deterpenated fractions of Origanum majorana L. essential oil on the final properties of chitosan/ β-chitin nanofibres nanocomposite films.
This paper deals with the preparation and characterization of active films of chitosan/chitin nanofiber loaded with Origanum deterpenated essential oil. Antifungal and cytotoxic activity on fibroblast cells was assessed.
Several issues should be addressed before being considered for publication.
-The abstract section is informative, but more quantitative values from results must be included.
- Introduction section.
Line 62…some references are needed.
Line 75… poly-β-(1/4)-N-acetyl-D-glucosamine… changed by (poly-β-(1-4)-D-glucosamine).
Line 79…Include some references of chitosan bioactivity.
I suggest introducing chitin and its nanofiber, considering it is not a well-established commercial product yet.
Methodology section.
Line 98-101. The methodology to obtain chitin nanofibers is wrong. It is for obtaining nanocrystals. Add the correct one.
Line 102-104…. Provide evidence of nanofibers characterization (13C CPMASS NMR and AFM). You can consider adding them as supplementary material.
Line 106-107… gives more details of chitin isolation for this new raw material.
Line 112 to 119…consider writing these sentences again to be more precise or present them as a Table.
Consider adding a title to each independent methodology.
Results section
The Table 2 results must be discussed before the one presented in Table 3.
Consider separating each analysis and add a title.
In Figure 4…I suggest replacing this figure and adding contact angle value at 60sec as a column in Table 2.
In Figure 6. I suggest showing the TGA curves as a single line of different colors and remove symbols.
Figure 7 must be presented in a better and clear way (no x-axis title, different bars types, bad pictures, etc)
General comment.
A deeper discussion of all your results is encouraging and connect them with the significance of the manuscript.
No comments about the smell of the films or the release of the EO from the matrix are presented. We encourage the author to discuss both topics to improve paper quality.
-Conclusions are supported by the results obtained, but it is large, and it must be improved. Some quantitative results can be presented in the abstract.
Author Response
Reviewer 3
Comments and Suggestions for Authors polymers-1204339
The effect of deterpenated fractions of Origanum majorana L. essential oil on the final properties of chitosan/ β-chitin nanofibres nanocomposite films.
This paper deals with the preparation and characterization of active films of chitosan/chitin nanofiber loaded with Origanum deterpenated essential oil. Antifungal and cytotoxic activity on fibroblast cells was assessed.
Several issues should be addressed before being considered for publication.
-The abstract section is informative, but more quantitative values from results must be included.
Thanks for the comment we have changed and improve the abstract including more results.
- Introduction section.
Line 62…some references are needed.
Line 75… poly-β-(1/4)-N-acetyl-D-glucosamine… changed by (poly-β-(1-4)-D-glucosamine).
Line 79…Include some references of chitosan bioactivity.
I suggest introducing chitin and its nanofiber, considering it is not a well-established commercial product yet.
Thank you for your comments, we have now included more references and changed all suggestions.
Methodology section.
Line 98-101. The methodology to obtain chitin nanofibers is wrong. It is for obtaining nanocrystals. Add the correct one.
Thank you for your comment. However, with acid hydrolysis, nanofibres morphology can be obtained from squid pen origin that means beta-chitin (please see the Figure S2, AFM image). Also, Shankar et al. 2015 obtained nanofibres of chitin (CH) by hydrolysis with 3N HCl (ratio of 1:20 w/v, CH: HCl) using reflux for 3h at 60 ºC. (S. Shankar, J. P. Reddy, J. W. Rhim, Hee, H. Y. Kim, Preparation, characterization, and antimicrobial activity of chitin nanofibrils reinforced carrageenan nanocomposite films, Carbohydr. Polym. 117 (2015), 468-475.)
Line 102-104…. Provide evidence of nanofibers characterization (13C CPMASS NMR and AFM). You can consider adding them as supplementary material.
We agree with the reviewer's comment, we have now added AFM and 13C-NMR figures in supplementary material.
Line 106-107… gives more details of chitin isolation for this new raw material.
Thanks for the recommendation we have now explained better the chitin isolation.
Line 112 to 119…consider writing these sentences again to be more precise or present them as a Table.
Thank you for your comments we have changed the sentence.
Consider adding a title to each independent methodology.
We entirely agree with the referee´s suggestion, we have now added the titles.
Results section
The Table 2 results must be discussed before the one presented in Table 3.
We agree with the reviewer's comment, we have now moved the results in Table 2 where they belong.
Consider separating each analysis and add a title.
We entirely agree with this suggestion, we have now added the titles.
In Figure 4…I suggest replacing this figure and adding contact angle value at 60sec as a column in Table 2.
Thank you for your suggestion. However, we believe that with this figure we can better study the kinetics of water absorption at the surface of the nanocomposite films from droplet deposition at t = 0 s and 120 s.
In Figure 6. I suggest showing the TGA curves as a single line of different colors and remove symbols.
We agree with the reviewer's comments. We have now removed the symbols of the Figure.
Figure 7 must be presented in a better and clear way (no x-axis title, different bars types, bad pictures, etc)
Thank you for your comment, we have now change the Figure and the images of the general aspect after 7 days of incubation have been transferred to supplementary material as Figure S4.
General comment.
A deeper discussion of all your results is encouraging and connect them with the significance of the manuscript.
Thank you for your comment, we have now improved the manuscript discussion.
No comments about the smell of the films or the release of the EO from the matrix are presented. We encourage the author to discuss both topics to improve paper quality.
We fully agree with the reviewer's suggestion. However, the smell of the films and the release of the oil was not our focus of this work. In this article, we focus only on the physicochemical characterisation and biological properties like cytotoxicity and antifungal activity of chitosan/β-chitin nanofibres nanocomposite films with the incorporation of Origanum majorana L. oil and its fractions. However, this could be an interesting aspect to consider in future work. Thanks.
-Conclusions are supported by the results obtained, but it is large, and it must be improved. Some quantitative results can be presented in the abstract.
We fully agree with the reviewer's suggestion. We have improved the conclusion of the manuscript.
Round 2
Reviewer 2 Report
I am congratulating the authors for their enthusiasm and sincerity in addressing the referee´s comments.
Author Response
Dear Reviewer,
we thank you for your help in improving our paper.
Kind regards,
Susana Fernandes
Reviewer 3 Report
Dear authors,
There are few issues to attend before been considered for publication.
In the Results section. Table 4. Mechanical properties.
The letters from statistical analysis are not easy to follow, you will need to check them again.
Why the mechanical properties of the sample CSNF-F2 do not follow a clear trend ?. Can you provide an explanation on this matter.
The X-axis titles in figure 7 and 8 are missing and the Y-axis in Figure 8 should start with a Capital letter
Figure 8 has poor resolution, please provide a higher resolution figure.
In figure 8, the letters from statistical analysis are not easy to follow, please explain in the text how the analysis was performed (daily comparison or among all treatments?). You will need to check the error bar of the last treatment (OM).
Author Response
Dear Reviewer,
many thanks for your help in improving our manuscript.
Please, find below our answer to your comments:
There are few issues to attend before been considered for publication.
In the Results section. Table 4. Mechanical properties.
The letters from statistical analysis are not easy to follow, you will need to check them again.
Thanks for the suggestion, we check the statistical analysis and we have now modified.
Why the mechanical properties of the sample CSNF-F2 do not follow a clear trend ?. Can you provide an explanation on this matter.
Thank you for the comment, we believe that the trend of CSNF-F2 film in the YM is due to its composition. The composition of the oil fraction of this film is 12.90 % of terpenic hydrocarbon and 85.06 % of oxygenated terpenic derivatives.
The X-axis titles in figure 7 and 8 are missing and the Y-axis in Figure 8 should start with a Capital letter
Thank you for your suggestion, we have now improved the figures.
Figure 8 has poor resolution, please provide a higher resolution figure.
Thank you for the suggestion, we have now improved the resolution of the figure.
In figure 8, the letters from statistical analysis are not easy to follow, please explain in the text how the analysis was performed (daily comparison or among all treatments?). You will need to check the error bar of the last treatment (OM).
Thank you for the comment, the statistics were performed between the different films with the same cell culture time. A clarification has been added to the graph and the statistics have been revised.
Kind regards,
Susana Fernandes